# Deep Unsupervised Feature Selection

## Abstract

Unsupervised feature selection involves finding a small number of highly informative features, in the absence of a specific supervised learning task. Selecting a small number of features is an important problem in many scientific domains with high-dimensional observations. Here, we propose the *restricted autoencoder* (RAE) framework for selecting features that can accurately reconstruct the rest of the features. We justify our approach through a novel proof that the reconstruction ability of a set of features bounds its performance in downstream supervised learning tasks. Based on this theory, we present a learning algorithm for RAEs that iteratively eliminates features using learned per-feature corruption rates. We apply the RAE framework to two high-dimensional biological datasets—single cell RNA sequencing and microarray gene expression data, which pose important problems in cell biology and precision medicine—and demonstrate that RAEs outperform nine baseline methods, often by a large margin.

## 1 Introduction

Many domains involve high-dimensional observations $X \in \mathbb{R}^d$, and it is often desired to select a small number of representative features *a priori* and observe only this subset. For natural spatio-temporal signals, compressive sensing provides a popular solution to this problem (Candès et al., 2006; Donoho et al., 2006). While prior knowledge of the structure of data is not available in the more general setting, a similar approach can be fruitful: unsupervised feature selection should select features that are informative in a general sense, and not just for a specific supervised learning task.

As a motivating example, we may be restricted to measuring the expression levels of only a small number of genes, and then use these measurements in a variety of future prediction tasks, such as disease subtype prediction, cell type classification, and so on. More generally, big data problems are impacting virtually all research, and the ability to observe an informative subset of the original features offers important advantages. It yields knowledge about the structure of the data, reduces storage requirements, makes downstream computation more efficient, and alleviates the curse of dimensionality. By contrast with feature *extraction* methods, such as PCA (Jolliffe, 2011), feature selection also reduces the burden of measuring all features.

The prior work on unsupervised feature selection is diverse, with methods designed to provide good clustering results (Cai et al., 2010; Dy & Brodley, 2004) to preserve the local structure of data (He et al., 2006; Zhao & Liu, 2007; Cai et al., 2010), and to eliminate redundancy (Farahat et al., 2011; Mitra et al., 2002). The abundance of existing approaches reflects a lack of consensus on the correct optimization objective for unsupervised feature selection. We argue that for settings where the selected features will be used in downstream prediction tasks, the optimal approach is to select a set of features that can accurately reconstruct all the remaining features.

The contributions of this paper are the following:

1. *We propose the "restricted autoencoder" (RAE) framework for selecting features based on their reconstruction ability.* The approach is justified by novel theoretical results, which show that the reconstruction ability of a set of features bounds its performance in downstream supervised learning tasks.

2. *We propose an approximate optimization algorithm for the RAE objective function, based on learning per-feature corruption rates.* The algorithm trains a RAE by iteratively eliminating features that are not important for performing accurate reconstruction.

3. *We apply our approach to two biological datasets, where finding a small number of informative features is an important scientific problem.* Experiments demonstrate that RAEs outperform nine baseline methods: they select features that achieve lower reconstruction error and perform better in downstream prediction tasks.

## 2    RELATED WORK

Early work on unsupervised feature selection was motivated by the observation that PCA (Pearson, 1901; Hotelling, 1933) yields principal components (PCs) that depend on all input features. In two early studies, Jolliffe (1972) developed heuristics to discard features while roughly preserving the results of PCA, and Krzanowski (1987) proposed assessing the quality of a feature subset through a rotation of its PCs. McCabe (1984) derived several criteria for defining an optimal subset of features, taking inspiration from the numerous criteria that are optimized by PCA.

Among the criteria proposed by McCabe (1984), one represents the mean squared error (MSE) when the rejected features are reconstructed by the selected features using a linear function. The reconstruction loss was revisited in several more recent works, with each addressing the challenge of combinatorial optimization. Farahat et al. derive an efficient algorithm for greedy feature selection (Farahat et al., 2011), while Masaeli et al. (2010) and Zhu et al. (2015) select features by sparsifying an auto-associative linear model. Several methods optimize a similar objective using per-feature leverage scores (Boutsidis et al., 2009a;b; 2014; Papailiopoulos et al., 2014).

The prior work that considers reconstruction ability has focused primarily on reconstruction with a linear function, although one study proposed using multivariate regression trees (Questier et al., 2005), and there have been nascent efforts to perform feature selection with autoencoders (Chandra & Sharma, 2015; Han et al., 2018). The differences between our work and these studies are i) we rigorously justify the reconstruction loss through its implications for downstream prediction tasks, ii) we explain the importance of considering reconstruction with a nonlinear function, and iii) we propose an algorithm that optimizes the objective function more effectively.

Other approaches for unsupervised feature selection find clusters of similar features (Mitra et al., 2002; Lu et al., 2007), select non-redundant features with a greedy algorithm based on PCA (Cui & Dy, 2008), use spectral information to preserve the local structure of data (He et al., 2006; Zhao & Liu, 2007; Zhao et al., 2010), retain local discriminative information (Yang et al., 2011), and attempt to preserve clustering structure (Dy & Brodley, 2004; Boutsidis et al., 2009a; Cai et al., 2010). These approaches are designed with other aims, not to select features for use in supervised learning tasks.

In Section 5, we implement many of these methods and compare them with our approach using multiple datasets and downstream experiments.

## 3    FEATURE SELECTION BASED ON IMPUTATION ABILITY

### 3.1    IMPUTATION LOSS

For a random variable $X \in \mathbb{R}^d$, feature selection algorithms determine a set $\mathcal{S} \subset \{1, 2, \ldots, d\}$ of selected indices, and a set $\mathcal{R} \equiv \{1, 2, \ldots, d\} \setminus \mathcal{S}$ of rejected indices. We use the notation $X^{\mathcal{S}} \in \mathbb{R}^{|\mathcal{S}|}$ and $X^{\mathcal{R}} \in \mathbb{R}^{|\mathcal{R}|}$ to denote selected and rejected features, respectively. For notational convenience we assume that all random variables have zero mean.

The goal of unsupervised feature selection is to select features $X^{\mathcal{S}}$ that are most representative of the full observation vector $X$. An approach that has received some interest in prior work is to measure how well $X^{\mathcal{S}}$ can reconstruct $X^{\mathcal{R}}$ (McCabe, 1984; Questier et al., 2005; Farahat et al., 2011; Masaeli et al., 2010; Papailiopoulos et al., 2014; Zhu et al., 2015; Han et al., 2018). It is intuitive to consider reconstruction ability, because if the rejected features can be reconstructed perfectly, then no information is lost when selecting a subset of features. To make the motivation for this approach precise, we derive a rigorous justification that has not been presented in prior work.

To quantify reconstruction ability, we define the *imputation loss* $\mathcal{L}(\mathcal{S})$ and the *linear imputation loss* $\mathcal{L}_{\text{linear}}(\mathcal{S})$. Both quantify how much information $X^{\mathcal{S}}$ contains about $X^{\mathcal{R}}$, and it is clear from their definitions that $\mathcal{L}(\mathcal{S}) \leq \mathcal{L}_{\text{linear}}(\mathcal{S})$.

**Definition 1** (Imputation loss). *The imputation loss $\mathcal{L}(\mathcal{S})$ and the linear imputation loss $\mathcal{L}_{\text{linear}}(\mathcal{S})$ quantify how well $X^{\mathcal{S}}$ can reconstruct $X^{\mathcal{R}}$ using an unrestricted function, and using a linear function, respectively. They are defined as:*

$$\mathcal{L}(\mathcal{S}) = \min_{h} \ \mathbb{E}[\ ||X^{\mathcal{R}} - h(X^{\mathcal{S}})||^2\ ] \tag{1}$$

$$\mathcal{L}_{\text{linear}}(\mathcal{S}) = \min_{W} \ \mathbb{E}[\ ||X^{\mathcal{R}} - WX^{\mathcal{S}}||^2\ ] \tag{2}$$

Either one could serve as a feature selection criterion, so we aim to address the following questions:

1. Why is it desirable to optimize for the reconstruction ability of selected features $X^{\mathcal{S}}$?
2. Is it preferable to measure reconstruction ability using $\mathcal{L}(\mathcal{S})$ or $\mathcal{L}_{\text{linear}}(\mathcal{S})$?

## 3.2 Implications of Imputation Loss for Downstream Tasks

In this section, we demonstrate through theoretical results that reconstruction ability is connected to performance in downstream supervised learning tasks. These results aim to characterize the usefulness of $X^{\mathcal{S}}$ for predicting a target variable $Y \in \mathbb{R}$. All proofs are in Appendix C.

To facilitate this analysis, we assume that all learned models are optimal; this assumption ignores practical problems such as overfitting and non-convex optimization, but makes it possible to shed light on the connection between reconstruction ability and predictive power. Our analysis focuses on the degradation in performance (henceforth called *performance loss*) when a model is fitted to $X^{\mathcal{S}}$ instead of $X$. Intuitively, performance in the downstream task must suffer when information is discarded. The question is, by how much?

Consider a fixed partitioning of features into $(X^{\mathcal{S}}, X^{\mathcal{R}})$. The first result addresses the situation when a *linear model* is used to predict $Y$ given $X^{\mathcal{S}}$. Theorem 1 states that the performance loss has an exact expression, which can be related to $\mathcal{L}_{\text{linear}}(\mathcal{S})$. For this result we define the notation $\Sigma_s = \text{Cov}(X^{\mathcal{S}})$, $\Sigma_r = \text{Cov}(X^{\mathcal{R}})$, $\Sigma_{sr} = \text{Cov}(X^{\mathcal{S}}, X^{\mathcal{R}})$, and $\Sigma_{r|s} = \Sigma_r - \Sigma_{sr}\Sigma_s^{-1}\Sigma_{sr}$.

**Theorem 1** (Performance loss with linear model). *Assume a prediction target $Y$ such that*

$$(b_*, c_*) = \arg\min_{b,c} \ \mathbb{E}\big[(Y - b^T X^{\mathcal{S}} - c^T X^{\mathcal{R}})^2\big]. \tag{3}$$

*Then, the performance loss for features $X^{\mathcal{S}}$ is:*

$$\min_{u} \ \mathbb{E}\big[(Y - u^T X^{\mathcal{S}})^2\big] - \min_{v} \ \mathbb{E}\big[(Y - v^T X)^2\big] = c_*^T \Sigma_{r|s} c_*. \tag{4}$$

The performance loss is a quadratic form based on the matrix $\Sigma_{r|s}$, so it is dependent on the eigenstructure of the matrix. To ensure strong performance in downstream tasks, it is therefore desirable to select $X^{\mathcal{S}}$ so that $\Sigma_{r|s}$ has small eigenvalues. The result is notable because it can be shown that the linear imputation loss is equal to the sum of the eigenvalues, i.e., $\mathcal{L}_{\text{linear}}(\mathcal{S}) = \text{Tr}(\Sigma_{r|s})$. Selecting $X^{\mathcal{S}}$ to minimize $\mathcal{L}_{\text{linear}}(\mathcal{S})$ therefore minimizes the performance loss with linear models. This perspective has not been presented in prior work, but lends support to several existing approaches (Masaeli et al., 2010; Farahat et al., 2011; Papailiopoulos et al., 2014; Zhu et al., 2015).

The second result addresses the situation where a *nonlinear model* is used to predict $Y$. Nonlinear models (e.g., neural networks, gradient boosting machines) are predominant in many machine learning applications, so this result is more relevant to contemporary practices. Theorem 2 states that the performance loss can be upper bounded using $\mathcal{L}(\mathcal{S})$.

The result requires a mild assumption about the Hölder continuity (a generalization of Lipschitz contintuity) of the conditional expectation function $\mathbb{E}[Y \mid X = x]$ that is not verifiable in practice. Nonetheless, it connects the performance of $X^{\mathcal{S}}$ in prediction tasks with its ability to impute $X^{\mathcal{R}}$.

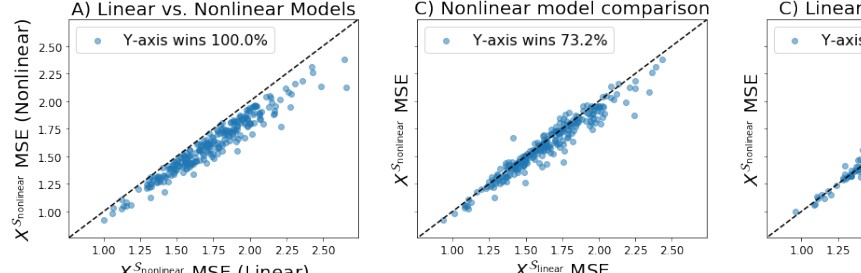

Figure 1: Feature set comparison on simulated prediction tasks. Each point represents a single task, and the legend shows the portion of tasks in which the method on y-axis outperforms the method on x-axis.

**Theorem 2** (Performance loss with nonlinear model). *Assume a prediction target $Y$ such that the conditional expectation $\mathbb{E}[Y \mid X = x]$ is $(C, \alpha)$-Hölder continuous with exponent $0 < \alpha \leq 1$, so that the following holds almost everywhere in the distribution of $X$:*

$$\big| \mathbb{E}[Y \mid X = a] - \mathbb{E}[Y \mid X = b] \big| \leq C \cdot ||a - b||_2^\alpha. \tag{5}$$

*Then, the performance loss for features $X^{\mathcal{S}}$ can be upper bounded:*

$$\min_{f_1} \mathbb{E}\big[(Y - f_1(X^{\mathcal{S}}))^2\big] - \min_{f_2} \mathbb{E}\big[(Y - f_2(X))^2\big] \leq C^2 \cdot \mathcal{L}(\mathcal{S})^\alpha. \tag{6}$$

The bound in Theorem 2 suggests that $X^{\mathcal{S}}$ should be selected to minimize $\mathcal{L}(\mathcal{S})$, because doing so reduces the upper bound on the performance loss. We appeal to the minimax principle to argue that the minimization of $\mathcal{L}(\mathcal{S})$, by effectively minimizing the worst-case performance loss, provides the most conservative way to select a small number of features (Von Neumann & Morgenstern, 1944).

While this result is easiest to prove for Hölder continuous tasks with the MSE loss function, the idea that good reconstruction ability leads to strong predictive power should apply more broadly. Our experiments in Section 5 confirm this empirically.

To make these results concrete, we present a simulation experiment involving two feature sets from the single-cell RNA sequencing data described in Section 5. The feature sets $X^{\mathcal{S}_{\text{nonlinear}}}$ and $X^{\mathcal{S}_{\text{linear}}}$ were chosen so that $\mathcal{L}(X^{\mathcal{S}_{\text{nonlinear}}}) \leq \mathcal{L}(X^{\mathcal{S}_{\text{linear}}})$, but $\mathcal{L}_{\text{linear}}(X^{\mathcal{S}_{\text{nonlinear}}}) \geq \mathcal{L}_{\text{linear}}(X^{\mathcal{S}_{\text{linear}}})$. We simulated 250 response variables with linear dependencies on subsets of the original features (see Section 5.4), and trained linear and nonlinear (neural network) predictive models for each task.

Figure 1 shows the relative performance of the two feature sets. The important observations from this experiment are that across a wide variety of tasks, nonlinear models perform better overall (Figure 1A), $X^{\mathcal{S}_{\text{nonlinear}}}$ usually performs better when using *nonlinear* models (Figure 1B), and $X^{\mathcal{S}_{\text{linear}}}$ usually performs better when using *linear* models (Figure 1C). These results provide empirical confirmation of both Theorem 1 and Theorem 2, and summarize our approach to feature selection.

To our knowledge, these results are novel in the space of unsupervised feature selection, despite the long history of the linear imputation loss (McCabe, 1984) and recent interest in the nonlinear imputation loss (Questier et al., 2005; Han et al., 2018). While it is tangential to unsupervised feature selection, we also show in Appendix A that the results can be stated more generally for unsupervised feature *extraction*.

Given these findings, and the predominance of nonlinear models in contemporary machine learning, we proceed with an approach to select $X^{\mathcal{S}}$ by minimizing the imputation loss as follows:

$$\mathcal{S}^* = \arg\min_{|\mathcal{S}|=k} \big\{ \min_f \mathbb{E}[\, ||X^{\mathcal{R}} - f(X^{\mathcal{S}})||^2 \,] \big\} \tag{7}$$

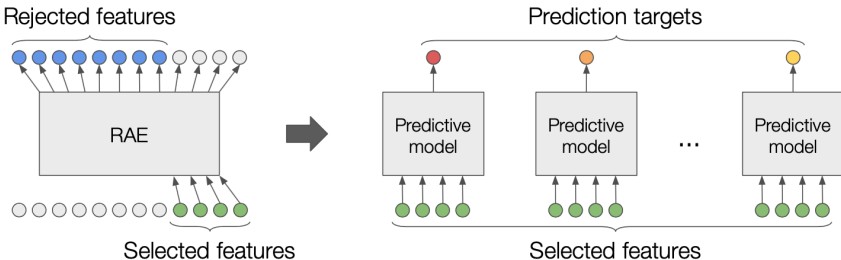

Figure 2: Features are selected by learning a restricted autoencoder (RAE), and can be applied in downstream prediction tasks.

## 4 RESTRICTED AUTOENCODERS FOR FEATURE SELECTION

### 4.1 RESTRICTED AUTOENCODERS

We aim to select $k$ features by solving the problem in Eq. 7. However, two obstacles make the problem difficult. First, $\mathcal{L}(\mathcal{S})$ is unknown in practice because we do not have access to the optimal imputation functions. Second, the combinatorial number of sets of size $k$ make an exhaustive search infeasible. We therefore propose the framework of *restricted autoencoders* (RAEs) to jointly optimize for $\mathcal{S}$ and $f$, and train a model to reconstruct the full observation vector while relying on a subset of the inputs. The approach is depicted in Figure 2. The concept of a RAE is straightforward, but it requires a non-trivial learning algorithm.

Sparsity inducing penalties (Feng & Simon, 2017; Tank et al., 2018) provide one option, but we found that these did not perform well on large datasets. Indeed, experiments in Section 5 show that the method of Han et al. (2018) fails to effectively optimize for reconstruction ability. Instead, we introduce a learning algorithm based on backwards elimination.

There are multiple ways to measure the sensitivity of a network to each input, and these can easily be adapted into feature ranking methods; highly ranked features are those that are most critical for performing accurate reconstruction. Instead of simply selecting the top ranked features, it may prove beneficial to reject a small number of features, and reassess the importance of the remaining ones. Following this logic, we present Algorithm 1 for learning a RAE by iteratively training a model, ranking features, and eliminating the lowest ranked features, in a procedure that is analogous to recursive feature elimination (Guyon et al., 2002).

To improve the running time of Algorithm 1, $h_\theta$ can be warm started using the model from the previous iteration. A multi-layer perceptron (MLP) is a natural choice because of the multitask prediction target; the model from the previous iteration can be modified by deleting columns from the first layer's weight matrix corresponding to eliminated features. In practice, we observe that this significantly reduces the number of training steps at each iteration.

---
**Algorithm 1:** Learning RAE

**inputs :** $dataset\ \{X_i\}_{i=1}^n$, $schedule$
$\mathcal{S} \leftarrow \{1, 2, \ldots, p\}$
**for** $k$ in $schedule$ **do**
    train $h_\theta : \mathbb{R}^{|\mathcal{S}|} \mapsto \mathbb{R}^d$ to predict $X$ given $X^{\mathcal{S}}$
    rank features in $\mathcal{S}$
    $\mathcal{S} \leftarrow \{k$ highest ranked features in $\mathcal{S}\}$
**end**
**return** $\mathcal{S}$

---

### 4.2 FEATURE RANKING METHODS

The purpose of the ranking step is to determine features that can be removed from $\mathcal{S}$ without impacting the model's accuracy. We consider two sensitivity measures, both of which are based on learning per-feature corruption rates. The first method stochastically sets inputs to zero using learned dropout rates $p_j$ for each feature $j \in \mathcal{S}$ (Chang et al., 2017). Similarly, the second method injects Gaussian noise using learned per-feature standard deviations $\sigma_j$. We refer to these methods as Bernoulli RAE and Gaussian RAE, due to the kind of noise they inject. Based on the logic that important features tolerate less corruption, we rank features according to $p_j$ or $\sigma_j$.

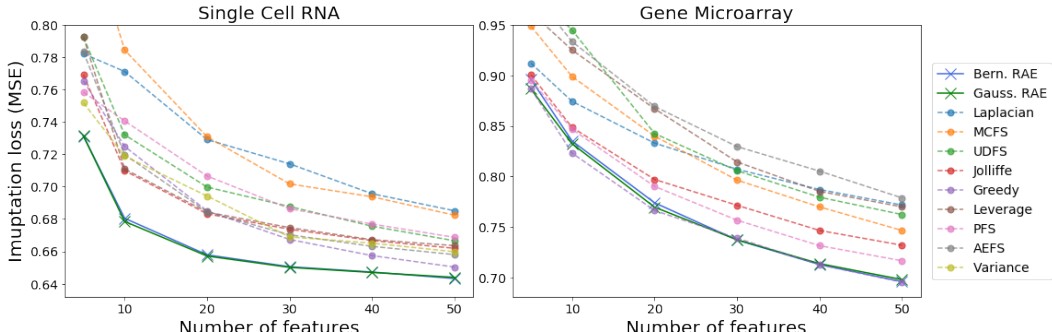

Figure 3: Imputation loss results. The MSE is normalized by the total variance of each dataset.

During training, both methods require penalty terms to encourage non-zero corruption rates. A hyperparameter $\lambda$ controls the tradeoff between accurate reconstruction and the amount of noise injected. The objective functions to be optimized at each iteration of Algorithm 1 are shown in Eqs. 8-9, and both are optimized using stochastic gradient methods and the reparameterization trick (Rezende et al., 2014; Maddison et al., 2016).

$$\min_{\theta,p} \; \mathbb{E}_{m\sim B(p)}\big[\, \mathbb{E}_X[(X - h_\theta(X^\mathcal{S} \odot m))^2]\,\big] - \lambda \sum_{j\in\mathcal{S}} \log p_j \qquad (8)$$

$$\min_{\theta,\sigma} \; \mathbb{E}_{z\sim N(0,\sigma^2)}\big[\, \mathbb{E}_X[(X - h_\theta(X^\mathcal{S} + z))^2]\,\big] + \lambda \sum_{j\in\mathcal{S}} \log(1 + \frac{1}{\sigma_j^2}) \qquad (9)$$

## 5 EXPERIMENTS

### 5.1 DATASETS AND BASELINE METHODS

We apply the RAE feature selection approach to two publicly available biological datasets: single-cell RNA sequencing data, and microarray gene expression data. The single-cell RNA sequencing dataset is from the Allen Brain Atlas, and contains expression counts for $n = 24,411$ cells and $d = 5,081$ genes (Tasic et al., 2018). The microarray data were collected from patients with breast cancer. Our dataset combines unlabeled samples from multiple Gene Expression Omnibus datasets ($n = 11,963$), as well as labeled samples from The Cancer Genome Atlas (TCGA) ($n = 590$), and we consider all genes that are present in both datasets ($d = 7,592$). We followed standard pre-processing techniques: we used log1p of the expression counts for the single-cell RNA sequencing data, and applied batch correction to the combined gene microarray datasets.

For both of these data domains, determining a small subset of informative features is an important problem. In precision medicine, a key goal is to identify a small set of expression markers for subtype classifications. As another example, fluorescent *in-situ* hybridization (FISH) methods (Raj et al., 2008; Chen et al., 2015) that measure the expression levels of pre-selected genes on intact tissue have become popular in brain sciences. However, such probes are typically limited to a handful of genes per hybridization round, and a problem of practical significance is to design the most informative FISH probes using as few genes as possible. The resulting spatial transcriptomic data can then be used to study multiple scientific problems (i.e., downstream tasks) relating to cell function, communication, and tissue organization.

We compare RAEs with nine baseline methods. Jolliffe B4 (Jolliffe, 1972) and principal feature selection (PFS, Cui & Dy (2008)) are both based on preserving the results of PCA. Greedy feature selection (GFS, Farahat et al. (2011)) and the leverage score method (Papailiopoulos et al., 2014) both optimize for reconstruction with a linear function. Autoencoder feature selection (AEFS, Han et al. (2018)) selects features by attempting to induce sparsity in a shallow MLP. Max variance simply selects features with the largest variance; it could not be applied to the microarray data,

Table 1: Classification accuracy using subsets of features

| # Features | Cell type classification | | | | | Cancer subtype classification | | | | |
|---|---|---|---|---|---|---|---|---|---|---|
| | 5 | 10 | 20 | 30 | 50 | 5 | 10 | 20 | 30 | 50 |
| Laplacian | 0.219 | 0.251 | 0.443 | 0.505 | 0.680 | 0.676 | 0.640 | **0.748** | 0.748 | 0.748 |
| MCFS | 0.111 | 0.278 | 0.532 | 0.622 | 0.713 | 0.532 | 0.514 | 0.613 | 0.685 | 0.685 |
| UDFS | 0.291 | 0.510 | 0.656 | 0.702 | 0.767 | 0.505 | 0.532 | 0.631 | 0.640 | 0.649 |
| PFS | 0.268 | 0.335 | 0.465 | 0.565 | 0.649 | 0.622 | 0.685 | 0.703 | 0.721 | 0.712 |
| AEFS | 0.320 | 0.574 | 0.759 | 0.806 | 0.847 | 0.523 | 0.486 | 0.550 | 0.640 | 0.604 |
| Variance | 0.447 | 0.541 | 0.741 | 0.793 | 0.822 | | | | | |
| Leverage | 0.463 | 0.634 | 0.780 | 0.816 | **0.852** | 0.523 | 0.568 | 0.613 | 0.658 | 0.649 |
| Jolliffe | 0.264 | 0.557 | 0.712 | 0.793 | 0.844 | 0.667 | 0.676 | 0.622 | 0.685 | 0.703 |
| Greedy | 0.203 | 0.367 | 0.580 | 0.691 | 0.820 | 0.657 | 0.673 | 0.684 | **0.750** | **0.753** |
| B. RAE | 0.484 | **0.674** | **0.789** | **0.822** | 0.845 | **0.679** | **0.687** | 0.701 | 0.721 | **0.753** |
| G. RAE | **0.487** | **0.667** | 0.771 | **0.822** | 0.846 | 0.645 | 0.678 | 0.686 | 0.694 | 0.740 |

because batch correction sets all features to have unit variance. Laplacian scores (He et al., 2006) and multi-cluster feature selection (MCFS, Cai et al. (2010)) both aim to preserve local structure in the data through spectral information. Unsupervised discriminative feature selection (UDFS, Yang et al. (2011)) aims to retain local discriminative information.

## 5.2 IMPUTATION ABILITY

We first demonstrate that RAEs select features that achieve a low imputation loss. Both datasets were split into training, validation and test sets, and we used only the unlabeled samples for the gene microarray data. For both datasets, we trained 20 RAEs by iteratively eliminating features using learnable Bernoulli and Gaussian noise. After features were selected, a separate imputation model was trained to predict only the rejected features, and the best features from the 20 trials were selected using a validation set. Similarly, imputation models were trained using features selected by all the baseline methods. The imputation loss was then estimated on the test set.

Hyperparameter choices were made on validation data, and are detailed in Appendix E. The RAEs are robust to both shallow and deep architectures, because the feature rankings are more important than the prediction accuracy. Iteratively eliminating features proved to be critical for both datasets, and we eliminated 20% of the remaining features at each iteration of Algorithm 1.

Figure 3 displays the results, showing the imputation loss for different numbers of selected features. RAEs achieve the best performance on both datasets. The gap is larger on the single-cell RNA sequencing data, where the RAEs perform significantly better than all baselines. RAEs and GFS achieve similar results on the microarray data, outperforming all other methods by a large margin.

GFS is a competitive baseline method, possibly because while it optimizes for linear reconstruction ability, it has the benefit of performing forward selection. Jolliffe and PFS are somewhat competitive, likely due to implicit connections between PCA and the linear imputation loss. Meanwhile, AEFS does not appear to effectively optimize for nonlinear reconstruction ability, and the non-reconstruction methods (Laplacian, MCFS, UDFS) naturally lead to poor reconstruction. In Appendix D we provide these results in a table. In Appendices F-G we demonstrate the importance of eliminating features iteratively, and analyze the variability between trials of RAEs.

## 5.3 REAL DOWNSTREAM PREDICTION TASKS

Next, we assess the performance of selected features in downstream prediction tasks. Both datasets have associated classification problems that, in certain settings, would need to be performed using a subset of features: cell type classification (150 types) for the single-cell RNA data, and cancer subtype classification (4 types) for the microarray data. For the single-cell data, we used the same dataset split; for cancer classification we split the labeled TGCA samples. MLPs were trained for each task, and the reported accuracy is the average performance of 10 models on the test data.

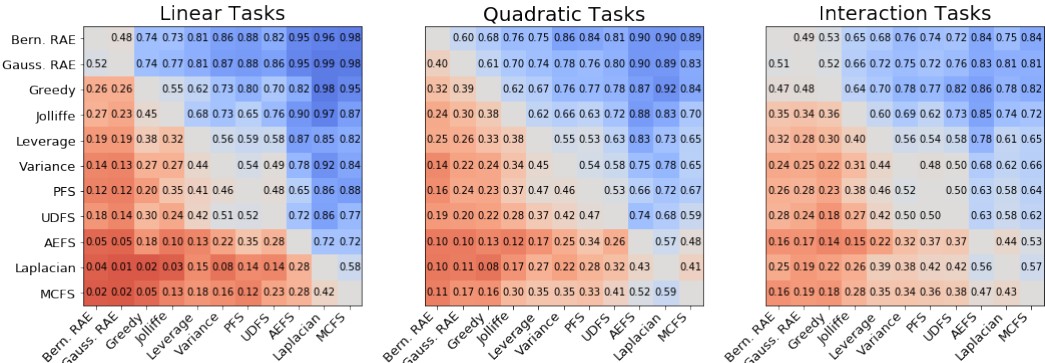

Figure 4: Results from the simulation study using scRNA data. Heatmaps provide pairwise comparisons, showing the portion of tasks in which features from the method on the y-axis achieved better performance than features from the method on the x-axis.

Table 1 displays the results for both datasets. Features selected by RAEs perform very well in both tasks, particularly when using a smaller number of features. Overall, RAEs achieve the best performance most of the time (7/10), and when they do not, they are still among the best. In Appendix D we present the same results in a plot that reveals a strong correlation between imputation ability and classification accuracy. We posit that features selected by RAEs perform well *not* because they were lucky to select "marker genes" for these tasks, but because they were optimized for reconstruction ability, and are therefore guaranteed to perform well in a variety of prediction problems.

## 5.4 LARGE-SCALE SIMULATION STUDY

The prediction problems in Section 5.3 provide one approach for verifying that RAEs select features with strong predictive power. However, it would be more informative to assess performance across a large number of prediction tasks. Since these two datasets do not offer a large number of prediction targets, our final experiment is a large-scale simulation study.

Using the single-cell RNA sequencing data, we simulate response variables with three kinds of dependencies on the original features. The response variables are simulated with either linear dependencies, quadratic dependencies, or involve interactions terms. There are 250 prediction targets of each type, and the simulation method is described in more detail in Appendix H. We then trained MLPs to predict the response variables using the features selected by each algorithm.

The full results are in Appendix H, and a representative summary is displayed in Figure 4, with heatmaps showing how often features from one method achieve better performance than features from another method. These results show that the features selected by a Bernoulli RAE and Gaussian RAE, which achieve lower imputation loss than all baselines, also achieve better performance in a majority of prediction tasks. Conversely, the features that perform most poorly (MCFS, Laplacian scores) are those with the worst reconstruction ability. The simulation study provides further empirical evidence for the theory in Section 3.2, underscoring the idea that selecting features for reconstruction ability leads to strong performance across a wide variety of prediction problems.

## 6 DISCUSSION

In this work we proposed a feature selection approach based on reconstruction ability, which we showed both theoretically and empirically has implications for the performance of features in downstream prediction tasks. We present the framework of *restricted autoencoders* (RAEs), and propose a learning algorithm based on iterative elimination, using learnable per-feature corruption rates to identify features that are unimportant to the model. Given the firm theoretical foundation for this feature selection approach, future work could focus on developing better algorithms for training RAEs, e.g., through forward selection.

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

# A    THEOREMS FOR UNSUPERVISED FEATURE EXTRACTION

In Section 3.2, we presented theoretical results to motivate the use of a reconstruction loss for unsupervised feature selection. Similar results can be stated for the more general problem of unsupervised feature *extraction*.

Consider an embedding function $g : \mathbb{R}^d \mapsto \mathbb{R}^k$ for a random variable $X \in \mathbb{R}^d$. The embedding function induces a distribution for a new random variable $Z \in \mathbb{R}^k$, which can be used in place of $X$ in downstream tasks. Here, we state two theorems for how well $Z$ should perform when predicting a target variable $Y \in \mathbb{R}$.

The first result addresses the situation where the embedding function is linear, i.e., $Z = WX$, and a linear model is used for the prediction task. For this result, we define the notation $\Sigma = \text{Cov}(X)$, $\Sigma_{ZX} = \text{Cov}(Z, X)$, $\Sigma_Z = \text{Cov}(Z)$. We restrict our attention to linear operations such that $Z$ is non-degenerate, so that $\Sigma_Z = W\Sigma W^T$ is invertible. As in the main text, we assume that all random variables have mean zero.

**Theorem 3** (Performance loss with linear model). *Assume a prediction target $Y$ such that*

$$v_* = \arg\min_v \mathbb{E}[(Y - v^T X)^2]. \tag{10}$$

*Then, the performance loss for a linear embedding variable $Z = WX$ is:*

$$\min_u \ \mathbb{E}\big[(Y - u^T Z)^2\big] - \min_v \mathbb{E}\big[(Y - v^T X)^2\big] = v_*^T(\Sigma - \Sigma_{ZX}^T \Sigma_Z^{-1} \Sigma_{ZX})v_* \tag{11}$$

To see the connection with reconstruction ability, we point out that the reconstruction ability of $Z$ is the trace of precisely the same matrix, i.e., $\min_B \ \mathbb{E}[\ ||X - BZ||^2\ ] = \text{Tr}(\Sigma - \Sigma_{ZX}^T \Sigma_Z^{-1} \Sigma_{ZX})$. Therefore, learning $Z$ to achieve a low reconstruction error can be understood as minimizing the performance loss. This is precisely what is done by PCA (Jolliffe, 2011), so this result supports the classic idea of principal components regression (Jolliffe, 1982).

The second result addresses the situation where a nonlinear model is used for the prediction task. Similarly to Theorem 2, the result requires an assumption about the Hölder-continuity of the conditional expectation function.

**Theorem 4** (Performance loss with nonlinear model). *Assume a prediction target $Y$ such that the conditional expectation function $\mathbb{E}[Y \mid X = x]$ is $(C, \alpha)$-Hölder continuous with exponent $0 < \alpha \le 1$, so that the following holds almost everywhere in the distribution of $X$:*

$$\big|\ \mathbb{E}[Y \mid X = a] - \mathbb{E}[Y \mid X = b]\ \big| \le C \cdot ||a - b||_2^\alpha. \tag{12}$$

*Then, the performance loss for an embedding variable $Z = g(X)$ can be bounded:*

$$\min_{f_1} \ \mathbb{E}\big[\big(Y - f_1(X^\mathcal{S})\big)^2\big] - \min_{f_2} \ \mathbb{E}\big[\big(Y - f_2(X)\big)^2\big] \le C^2 \cdot \big(\min_h \ \mathbb{E}[\ ||X - h(Z)||^2\ ]\big)^\alpha \tag{13}$$

This second result provides support for autoencoders (AEs, Hinton & Salakhutdinov (2006)), which jointly learn an encoder $g$ and decoder $h$ to optimize the reconstruction loss that shows up in the bound of Eq. 13. The connection with feature selection is that $g$ is a very particular linear function.

# B    PROOFS OF THEOREMS FOR UNSUPERVISED FEATURE EXTRACTION

We begin by proving the results stated in Appendix A. In Appendix C we show that these lead to straightforward proofs for the results in the main text.

*Proof of Theorem 3.* We introduce the notation $\Sigma_{XY} = \text{Cov}(X, Y) \in \mathbb{R}^d$, and calculate the MSE arising from the optimal linear model fit to $X$.

$$
\begin{aligned}
v_* &= \arg\min_v \; \mathbb{E}[(Y - v^T X)^2] \\
&= \Sigma^{-1}\Sigma_{XY} \\
\mathbb{E}[(Y - v_*^T X)^2] &= \mathbb{E}[(Y - \Sigma_{XY}^T \Sigma^{-1} X)^2] \\
&= \text{Var}(Y) - \Sigma_{XY}^T \Sigma^{-1} \Sigma_{XY}
\end{aligned}
$$

Similarly, we calculate the MSE from the model fit to $Z$.

$$
\begin{aligned}
u_* &= \arg\min_u \; \mathbb{E}[(Y - u^T Z)^2] \\
&= (W\Sigma W^T)^{-1} W \Sigma_{XY} \\
\mathbb{E}[(Y - u_*^T Z)^2] &= \mathbb{E}[(Y - \Sigma_{XT}^T W^T (W\Sigma W^T)^{-1} Z)^2] \\
&= \text{Var}(Y) - \Sigma_{XY}^T W^T (W\Sigma W^T)^{-1} W \Sigma_{XY}
\end{aligned}
$$

Taking the difference between the two terms, we get the desired result:

$$
\begin{aligned}
\min_u \mathbb{E}\big[(Y - u^T Z)^2\big] - \min_v \mathbb{E}\big[(Y - v^T X)^2\big] &= \Sigma_{XY}^T \Sigma^{-1} \Sigma_{XY} - \Sigma_{XY}^T W^T (W\Sigma W^T)^{-1} W \Sigma_{XY} \\
&= v_*^T \Sigma v_* - v_*^T \Sigma W^T (W\Sigma W^T)^{-1} W \Sigma v_* \\
&= v_*^T \Sigma v_* - v_*^T \Sigma_{ZX}^T \Sigma_Z^{-1} \Sigma_{ZX} v_* \\
&= v_*^T (\Sigma - \Sigma_{ZX}^T \Sigma_Z^{-1} \Sigma_{ZX}) v_*
\end{aligned}
$$

$\square$

*Proof of Theorem 4.* Consider the terms on the left side of the inequality. Both can be understood in terms of conditional variance.

$$
\begin{aligned}
\min_{f_1} \mathbb{E}\big[\big(Y - f_1(Z)\big)^2\big] &= \min_{f_1} \mathbb{E}\big[\big(Y - \mathbb{E}[Y|Z]\big)^2\big] + \mathbb{E}\big[\big(f_1(Z) - \mathbb{E}[Y|Z]\big)^2\big] \\
&= \mathbb{E}[(Y - \mathbb{E}[Y|Z])^2] \\
&= \mathbb{E}[\text{Var}(Y|Z)] \\
\min_{f_2} \mathbb{E}\big[\big(Y - f_2(X)\big)^2\big] &= \min_{f_2} \mathbb{E}\big[\big(Y - \mathbb{E}[Y|X]\big)^2\big] + \mathbb{E}\big[\big(f_2(X) - \mathbb{E}[Y|X]\big)^2\big] \\
&= \mathbb{E}[(Y - \mathbb{E}[Y|X])^2] \\
&= \mathbb{E}[\text{Var}(Y|X)] \quad\quad\quad\quad\quad\quad (14)
\end{aligned}
$$

Now, consider the first term in more detail. The following can be shown by using the Hölder continuity of the conditional expectation function, and applying the law of total variance and Jensen's inequality.

$$\text{Var}(Y|Z=z) = \mathbb{E}_{X|Z=z}[\text{Var}(Y|Z=z,X)] + \text{Var}_{X|Z=z}(\mathbb{E}[Y|Z=z,X])$$

$$\text{Var}_{X|Z=z}(\mathbb{E}[Y|Z=z,X]) = \text{Var}_{X|Z=z}\big(\mathbb{E}\big[Y|Z=z,X\big] - \mathbb{E}\big[Y|X=\mathbb{E}[X|Z=z]\big]\big)$$

$$\leq \mathbb{E}_{X|Z=z}\big[\big(\mathbb{E}\big[Y|Z=z,X\big] - \mathbb{E}\big[Y|X=\mathbb{E}[X|Z=z]\big]\big)^2\big]$$

$$= \mathbb{E}_{X|Z=z}\big[\big(\mathbb{E}\big[Y|X\big] - \mathbb{E}\big[Y|X=\mathbb{E}[X|Z=z]\big]\big)^2\big]$$

$$\leq C^2 \cdot \mathbb{E}_{X|Z=z}[\,||X - \mathbb{E}[X|Z=z]||^{2\alpha}\,]$$

$$\leq C^2 \cdot \mathbb{E}[\,||X - \mathbb{E}[X|Z=z]||^2\,]^\alpha$$

$$= C^2 \cdot \text{Tr}\big(\text{Cov}(X|Z=z)\big)^\alpha$$

$$\min_{f_1} \mathbb{E}[(Y - f_1(Z))^2] = \mathbb{E}[\text{Var}(Y|Z=z)]$$

$$= \mathbb{E}_{X,Z}[\text{Var}(Y|Z,X)] + \mathbb{E}\big[C^2 \cdot \text{Tr}\big(\text{Cov}(X|Z)\big)^\alpha\big]$$

$$\leq \mathbb{E}_{X,Z}[\text{Var}(Y|Z,X)] + C^2 \cdot \text{Tr}\big(\mathbb{E}[\,\text{Cov}(X|Z)\,]\big)^\alpha$$

$$= \mathbb{E}[\text{Var}(Y|X)] + C^2 \cdot \text{Tr}\big(\mathbb{E}[\,\text{Cov}(X|Z)\,]\big)^\alpha \tag{15}$$

Then, combine Eqs. 14-15 to obtain the following bound:

$$\min_{f_1} \mathbb{E}\big[\big(Y - f_1(Z)\big)^2\big] - \min_{f_2} \mathbb{E}\big[\big(Y - f_1(X)\big)^2\big] \leq C^2 \cdot \text{Tr}\big(\mathbb{E}[\,\text{Cov}(X|Z)\,]\big)^\alpha \tag{16}$$

Finally, consider the term that appears in the right side of Eq. 16. It can also be understood in terms of conditional covariance. Substituting this value into the above inequality completes the proof.

$$\min_h \mathbb{E}[\,||X - h(Z)||^2\,] = \min_h \mathbb{E}[\,||X - \mathbb{E}[X|Z]||^2\,] + \mathbb{E}[\,||h(Z) - \mathbb{E}[X|Z]||^2\,]$$

$$= \mathbb{E}[\,||X - \mathbb{E}[X|Z]||^2\,]$$

$$= \mathbb{E}\big[\,\text{Tr}\big((X - \mathbb{E}[X|Z])(X - \mathbb{E}[X|Z])^T\big)\big]$$

$$= \text{Tr}\big(\mathbb{E}[\,\text{Cov}(X|Z)\,]\big)$$

$$\square$$

## C   PROOFS OF THEOREMS FOR UNSUPERVISED FEATURE SELECTION

Now, we proceed with proofs of the results presented in Section 3.2 of the main text. They follow in a straightforward manner from proofs in Appendix B.

*Proof of Theorem 1.* Feature selection is an embedding with a particular kind of linear function. It can be seen that $X^{\mathcal{S}} = Z = WX$ for $W \in \mathbb{R}^{k \times p}$ with a single 1 in each row, corresponding to the selected features, and zeros everywhere else. Without loss of generality, we can assume that the features are ordered such that the full covariance matrix can be partitioned into blocks corresponding to the selected features $\mathcal{S}$ and rejected features $\mathcal{R}$.

$$\Sigma = \begin{pmatrix} \Sigma_s & \Sigma_{sr} \\ \Sigma_{sr}^T & \Sigma_r \end{pmatrix}$$

Then, the following can be seen:

$$\Sigma - \Sigma_{ZX}^T \Sigma_Z^{-1} \Sigma_{ZX} = \Sigma - (W\Sigma)^T (W\Sigma W^T)^{-1} (W\Sigma)$$
$$= \Sigma - \begin{pmatrix} \Sigma_s \\ \Sigma_{sr}^T \end{pmatrix} \Sigma_s^{-1} \begin{pmatrix} \Sigma_s & \Sigma_{sr} \end{pmatrix}$$
$$= \Sigma - \begin{pmatrix} \Sigma_s & \Sigma_{sr} \\ \Sigma_{sr}^T & \Sigma_{sr}^T \Sigma_s^{-1} \Sigma_{sr} \end{pmatrix}$$
$$= \begin{pmatrix} 0 & 0 \\ 0 & \Sigma_{r|s} \end{pmatrix}$$

The expression for the performance loss then follows from Theorem 3.

$$\min_u \mathbb{E}\big[(Y - u^T X^{\mathcal{S}})^2\big] - \min_v \mathbb{E}\big[(Y - v^T X)^2\big] = \begin{pmatrix} b_* \\ c_* \end{pmatrix}^T \begin{pmatrix} 0 & 0 \\ 0 & \Sigma_{r|s} \end{pmatrix} \begin{pmatrix} b_* \\ c_* \end{pmatrix}$$
$$= c_*^T \Sigma_{r|s} c_*$$

$\square$

*Proof of Theorem 2.* The bound involving the imputation loss $\mathcal{L}(\mathcal{S})$ follows directly from Theorem 4, upon the observation that $\mathcal{L}(\mathcal{S})$ is precisely the term in the upper bound.

$$\min_h \mathbb{E}[\,||X - h(Z)||^2\,] = \min_h \mathbb{E}[\,||X - h(X^{\mathcal{S}})||^2\,]$$
$$= \min_h \mathbb{E}[\,||X^{\mathcal{R}} - h(X^{\mathcal{S}})||^2\,]$$
$$= \mathcal{L}(\mathcal{S})$$

$\square$

## D    IMPUTATION ABILITY AND DOWNSTREAM CLASSIFICATION RESULTS

In this section, we present results from Section 5.2 and Section 5.3 in alternate form. Table 2 shows the imputation loss for the single-cell RNA sequencing data, and Table 3 shows the results for the gene microarray data. As in the main text, the MSE is normalized to reflect the portion of total variance. It is clear that RAEs nearly always select features that achieve the best imputation loss.

Figure 5 shows the results from the downstream classification tasks in graphical form, for 10 features. The plot shows a very clear linear trend between imputatation ability and classification accuracy, further underscoring the idea that features selected by RAEs perform well precisely because they have good reconstruction ability.

## E    HYPERPARAMETER TUNING

In this section we detail all hyperparameter choices that were made in this work. In general, we followed best practices of tuning hyperparameters on training and validation data, and reserved the test set for getting a final estimate of a model's performance. The validation set was also used to perform early stopping: we trained all models until they failed to achieve a new best value of the loss function for several consecutive epochs.

Table 4 shows hyperparameter choices that were made for training RAEs with either Bernoulli or Gaussian noise (Section 5.2). The "schedule" indicates the portion of remaining features that were eliminated at each iteration of Algorithm 1. Eliminating features iteratively proved to be critical for these high-dimensional datasets (see Appendix G). For the penalty hyperparameter $\lambda$ that is used in both methods, * indicates that $\lambda$ was annealed at each iteration. After the model converged, features were eliminated if the average dropout rate $p_j$ exceeded 0.5. Otherwise, $\lambda$ was doubled and the

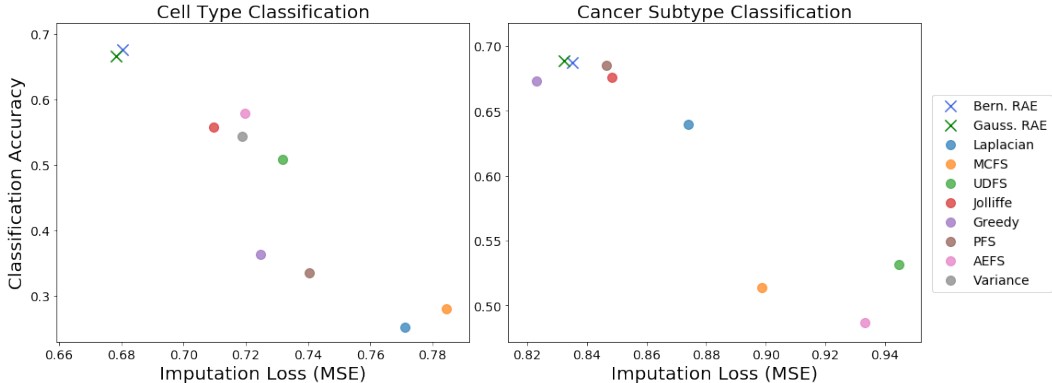

Figure 5: Downstream prediction accuracy versus imputation loss for subsets of 10 features selected by each method. Results across both datasets reveal a relationship between reconstruction ability and the predictive power of subsets of features.

Table 2: Single-cell RNA sequencing imputation loss (MSE)

| # Features | 5 | 10 | 20 | 30 | 40 | 50 |
|---|---|---|---|---|---|---|
| Laplacian | 0.782 | 0.771 | 0.729 | 0.714 | 0.696 | 0.685 |
| MCFS | 0.869 | 0.784 | 0.731 | 0.702 | 0.694 | 0.682 |
| UDFS | 0.792 | 0.732 | 0.700 | 0.688 | 0.675 | 0.666 |
| PFS | 0.758 | 0.741 | 0.706 | 0.686 | 0.677 | 0.669 |
| AEFS | 0.783 | 0.720 | 0.684 | 0.670 | 0.663 | 0.658 |
| Variance | 0.752 | 0.719 | 0.694 | 0.669 | 0.665 | 0.660 |
| Leverage | 0.793 | 0.711 | 0.684 | 0.675 | 0.667 | 0.663 |
| Jolliffe | 0.769 | 0.710 | 0.683 | 0.674 | 0.667 | 0.662 |
| Greedy | 0.765 | 0.725 | 0.685 | 0.667 | 0.657 | 0.650 |
| Bern. RAE | **0.731** | 0.680 | 0.658 | **0.650** | **0.647** | **0.643** |
| Gauss. RAE | **0.731** | **0.678** | **0.657** | **0.650** | **0.647** | 0.644 |

Table 3: Gene microarray imputation loss (MSE)

| # Features | 5 | 10 | 20 | 30 | 40 | 50 |
|---|---|---|---|---|---|---|
| Laplacian | 0.912 | 0.874 | 0.833 | 0.807 | 0.787 | 0.772 |
| MCFS | 0.949 | 0.899 | 0.840 | 0.796 | 0.770 | 0.747 |
| UDFS | 0.985 | 0.945 | 0.842 | 0.806 | 0.780 | 0.762 |
| PFS | 0.895 | 0.847 | 0.790 | 0.757 | 0.732 | 0.717 |
| AEFS | 0.984 | 0.933 | 0.869 | 0.830 | 0.805 | 0.779 |
| Leverage | 0.962 | 0.925 | 0.867 | 0.814 | 0.785 | 0.770 |
| Jolliffe | 0.900 | 0.848 | 0.797 | 0.772 | 0.747 | 0.732 |
| Greedy | **0.887** | **0.823** | **0.766** | 0.739 | **0.713** | **0.696** |
| Bern. RAE | 0.896 | 0.835 | 0.774 | **0.737** | **0.713** | **0.696** |
| Gauss. RAE | **0.887** | 0.832 | 0.770 | **0.737** | 0.714 | 0.698 |

Table 4: Restricted autoencoder hyperparameters

|  | Single-cell RNA | | Gene microarray | |
| --- | --- | --- | --- | --- |
|  | Bernoulli | Gaussian | Bernoulli | Gaussian |
| Optimizer | Adam | Adam | Adam | Adam |
| Learning rate | $10^{-3}$ | $10^{-3}$ | $10^{-3}$ | $10^{-3}$ |
| Minibatch size | 256 | 256 | 32 | 32 |
| Architecture | $[100] \times 4$ | $[100] \times 4$ | $[100] \times 2$ | $[100] \times 2$ |
| Activations | ELU | ELU | ELU | ELU |
| Schedule | 20% | 20% | 20% | 20% |
| $\lambda$ | 10.0* | 0.1 | 10.0* | 10.0 |

Table 5: Imputation model hyperparameters

|  | Single-cell RNA | Gene microarray |
| --- | --- | --- |
| Optimizer | Adam | Adam |
| Learning rate | $10^{-3}$ | $10^{-3}$ |
| Minibatch size | 264 | 256 |
| Architecture | $[100] \times 4$ | $[500] \times 2$ |
| Activations | ELU | ELU |

model was retrained. Overall, both methods were robust to most hyperparameters, except for the penalty parameter $\lambda$ and the schedule.

Table 5 shows hyperparameters that were used to train imputation models given the features selected by each algorithm. Table 6 shows hyperparameters that were used to train models for downstream tasks, including both real classification problems (Section 5.3) and simulated tasks (Section 5.4).

Finally, several baseline methods required hyperparameter choices. For Laplacian scores and MCFS, we calculated the 0-1 similarity matrix between data points using 5 nearest neighbors. For the leverage scores method, we calculated leverage scores using a latent dimension of 25. For AEFS, we used a MLP with a single hidden layer of size 100, and tuned the regularization hyperparameter on validation data.

## F  VARIANCE WITHIN TRIALS

Unlike many of the baseline methods we considered, RAEs do not select features deterministically. Variability arises from the model initialization and optimization, and manifests itself in the selection of different features in each trial. That variability is beneficial, in the sense that the best features can be selected by measuring their performance on validation data. Here, we analyze the amount of variability incurred by RAEs through several experiments on the single-cell RNA sequencing data.

In Figure 6 we plot the mean and standard deviation of the imputation loss across 20 trials. For context, we include two baseline methods: Jolliffe, which is somewhat competitive, and Laplacian

Table 6: Downstream prediction model hyperparameters

|  | Cell type | Cancer subtype | Simulated tasks |
| --- | --- | --- | --- |
| Loss function | Cross entropy | Cross entropy | MSE |
| Optimizer | Adam | Adam | Adam |
| Learning rate | $10^{-3}$ | $10^{-3}$ | $10^{-3}$ |
| Minibatch size | 64 | 32 | 32 |
| Architecture | $[100] \times 2$ | $[16] \times 1$ | $[50] \times 1$ |
| Activations | ELU | ELU | ELU |

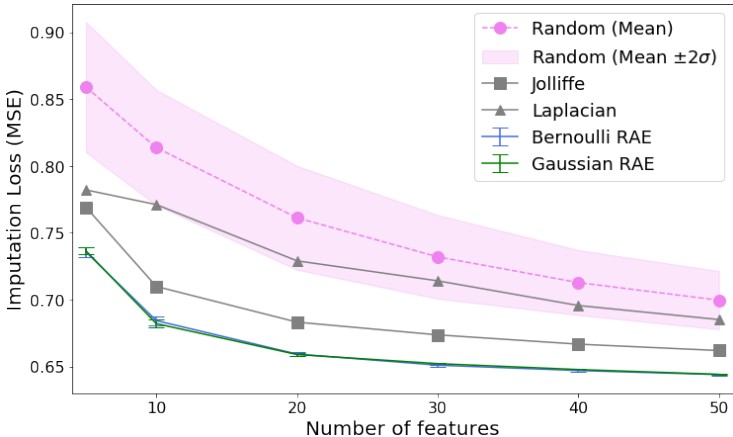

Figure 6: Variance of imputation loss achieved by RAEs, and comparison with randomly selected features.

Table 7: Stability analysis

|  | 10 Features | | 30 Features | | 50 Features | |
| --- | --- | --- | --- | --- | --- | --- |
|  | Jaccard | Adj. Rand | Jaccard | Adj. Rand | Jaccard | Adj. Rand |
| Bernoulli RAE | 0.722 | 0.833 | 0.722 | 0.834 | 0.718 | 0.830 |
| Gaussian RAE | 0.736 | 0.843 | 0.720 | 0.833 | 0.832 | 0.905 |

scores, which is among the least competitive baselines. We also include the results from 100 randomly selected sets of features. The results indicate that the amount of variance in the imputation loss from RAEs is negligible, compared to the gap with baseline methods. They also indicate that selecting features at random naturally performs much worse, but that it is easier to pick good features when the set is larger.

In Table 7 we quantify the consistency in the features that are selected within 20 trials. We calculate the average Jaccard index, and the average adjusted Rand index between each pair of trials. The results indicate that both methods are quite consistent, but that there is always variability.

To observe the variability in selected features across the two RAE methods, Bernoulli RAE and Gaussian RAE, Figure 7 shows the number of times each gene was selected among the 10 features, within 20 trials. Both methods are very consistent with 8 of their features, selecting them over 15 times within 20 trials. The two methods even tend to select 7 of the same features in most trials. These results show that the methods have a degree of consistency in the features they select. We leave for future work an analysis of the function of these genes for single-cell data.

## G   IMPORTANCE OF ITERATIVE ELIMINATION

To illustrate the importance of iterative elimination of features, as in Algorithm 1, we conduct an ablation experiment where we remove this aspect of the method. On both datasets, we learn RAEs by training an AE with Bernoulli or Gaussian noise a single time, ranking the features, and simply selecting the top ranked features. For a fair comparison, we chose the best results from 20 trials.

Figure 8 shows a comparison with the results in the main text, with the additional context of two baseline methods: Jolliffe, which is somewhat competitive, and Laplacian scores, which is among the least competitive baselines. The results demonstrate that iterative elimination is key to performing effective feature selection, especially for small numbers of features.

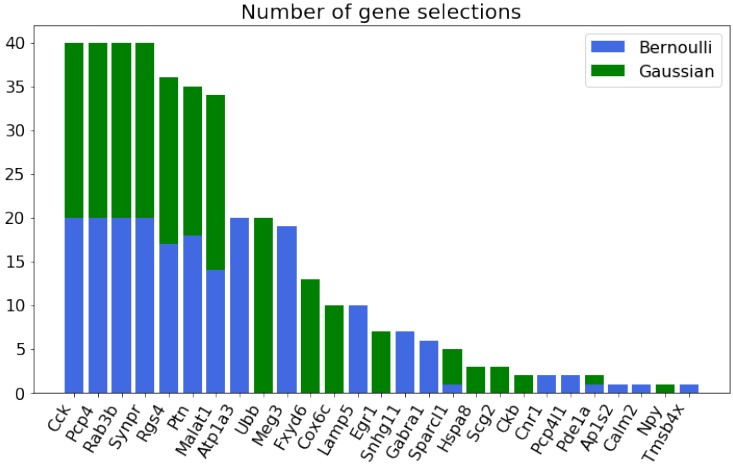

Figure 7: Bar chart of gene selections on single-cell RNA sequencing data. The stacked bar chart shows how many times each gene was among the 10 genes selected by each method, within 20 trials.

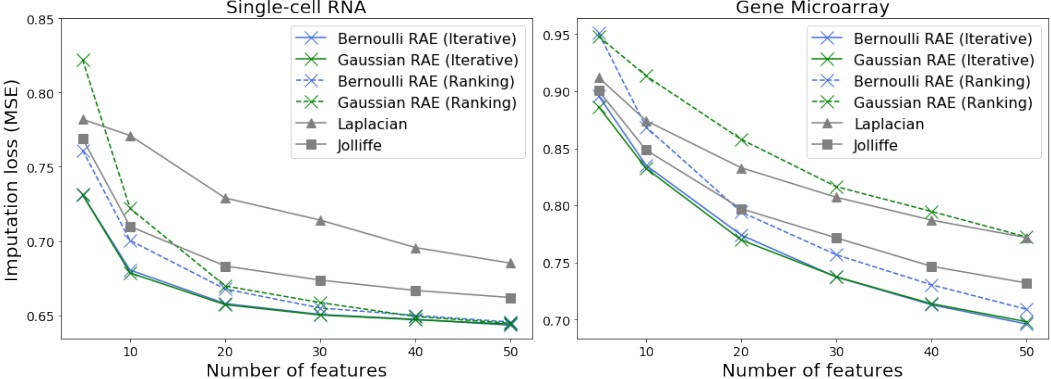

Figure 8: Comparison of RAEs learned through iterative elimination, and a single ranking step.

## H    SIMULATION STUDY

For the simulation study in Section 5.4, we simulated 250 response variables for each of three kinds of dependencies: linear, quadratic, and interaction effects. The precise method for simulating response variables was as follows.

For each the linear task, we selected 50 genes uniformly at random, then simulated a coefficient vector from $\mathcal{N}(0, I_{50})$. The response variables were then calculated for every observation vector in the dataset, with no additional noise added. The simulation method was similar for the quadratic and interaction effect tasks. For the quadratic tasks, we selected 25 genes to have linear dependencies, and 25 genes to have quadratic dependencies. For the interaction tasks, we selected 25 genes to have linear dependencies, and then 25 pairs of genes (selected uniformly at random), where the response variable depended linearly on the product of the two genes' expression levels.

The full results of the simulation study are shown in Figure 9, including results for sets of 10, 20, and 30 genes. The results show that the features selected by the Bernoulli RAE and Gaussian RAE outperform other features in a large portion of tasks. The most plausible explanation for these features' consistent ability to make better predictions is their superior reconstruction ability.

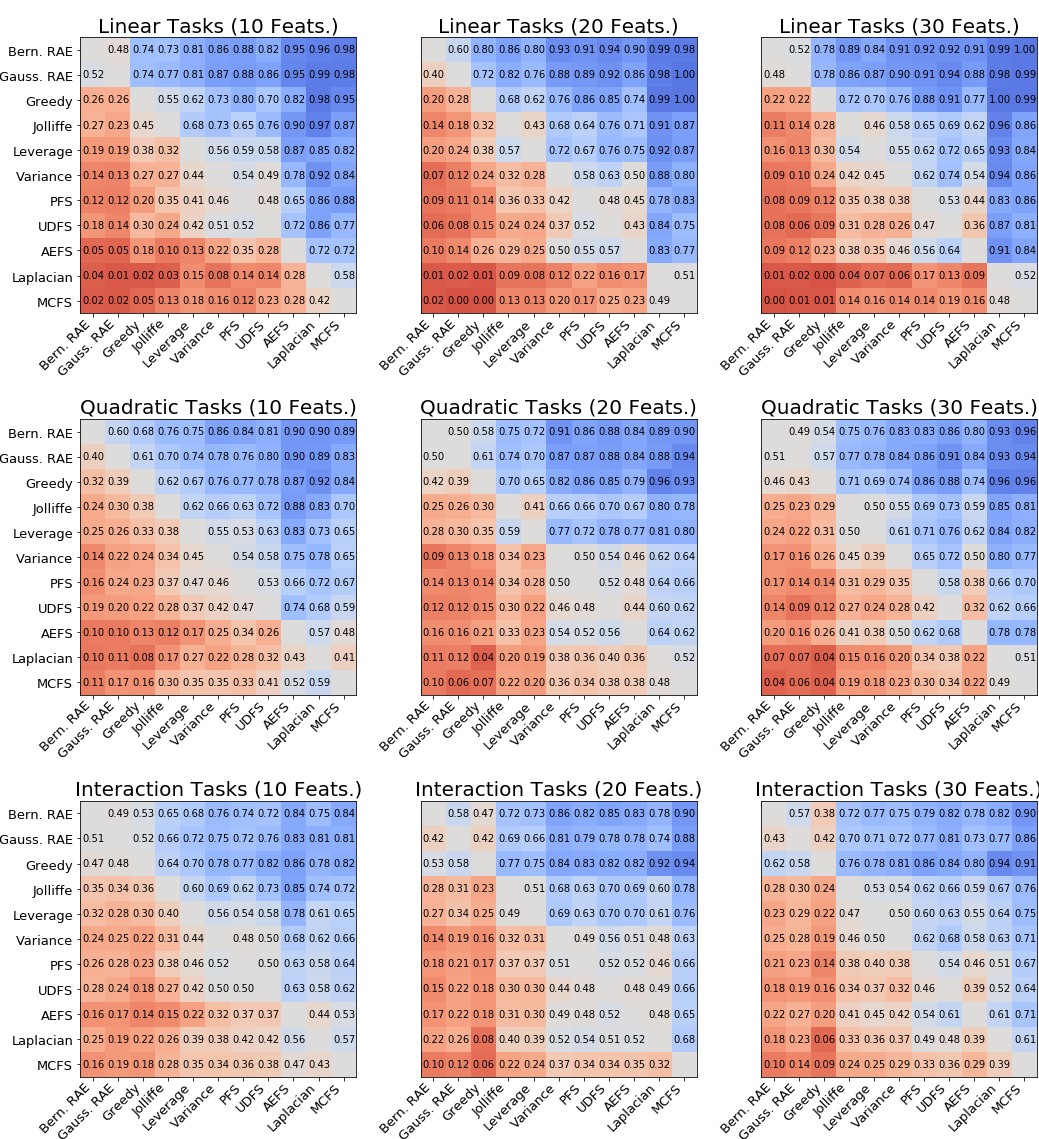

Figure 9: Results from the simulation study using single-cell RNA sequencing data. 250 response variables were simulated with three kinds of dependencies on the original features (750 total tasks) and were predicted using subsets of features. Heatmaps show pairwise comparisons between feature sets, displaying the portion of tasks in which features from the method on the y-axis outperformed those on the x-axis.

