# OpenReview forum: "Deep unsupervised feature selection"
_ICLR.cc/2020/Conference — Reject_

### Official Review · AnonReviewer1 · 2019-10-23
**Official Blind Review #1**

**Rating:** 3

**Review:**

Authors of this paper propose the restricted autoencoder (RAE) framework for selecting features that can accurately reconstruct the rest of features. Authors justify the proposed method via the proof that the reconstruction ability of a set of features bounds its performance in downstream supervised learning tasks. The algorithm that iteratively eliminates features using learned per-feature corruption rates is proposed.

The fundamental of this paper is built on the argument that the optimal approach is to select a set of features that can accurately reconstruct all the remaining features for the settings where they will be used in downstream prediction tasks. Authors studied the performance losses of linear and nonlinear models by using the defined imputation losses.  Some concerns are listed as follows:
1. the theorem based on strong assumptions that all learned models are optimal. The applicability of the theoretical results to general prediction model is still questionable.
2. only the prediction problems of least square (linear or nonlinear) are studied. It is not equivalent to the downstream supervised learning tasks. It is just a special case study.
3. It is unclear how to get the conclusion from Theorem 1 that the linear imputation loss is equal to the sum of eigenvalues. Please clarify it in details.

The RFE-like algorithm is used to solve (7). However, the sensitivity measures used in Algorithm 1 seems to take the different optimization problems since additional regularization terms are added. This is different from RFE where a single SVM optimization problem is used and the ranking score is solely based on the learned SVM classifier. The discussion on the inconsistency of learning h_{\theta} and the sensitivity measures could be interesting.

In the experiments, authors did not mention the parameter settings of all compared methods. It is known that the unsupervised feature selection methods incorporate priors with usually various parameters. For fair comparisons, it is better to report the properly tuned results since these parameters are often data-dependent.


**Experience Assessment:**

I have published one or two papers in this area.

**Review Assessment: Checking Correctness Of Derivations And Theory:**

I assessed the sensibility of the derivations and theory.

**Review Assessment: Checking Correctness Of Experiments:**

I carefully checked the experiments.

**Review Assessment: Thoroughness In Paper Reading:**

I read the paper at least twice and used my best judgement in assessing the paper.

---

### Official Review · AnonReviewer2 · 2019-10-25
**Official Blind Review #2**

**Rating:** 1

**Review:**

The paper proposes an unsupervised feature selection method by minimizing reconstruction error with restricted autoencoders. The proposed method employs iterative elimination to select features with learned per-feature corruption rates.

1) The novelty of the paper is very limited: using reconstruction for unsupervised feature selection has been explore in many papers, such as [1][2][3]. So the proposed method is a bit incremental.

2) Time complexity of the proposed method should be discussed.

3) Experimental results are not convincing at all due to the following reasons:
  a) Only one baseline (out of 9) was proposed after 2011. Outperforming these decade old approaches is not very difficult. There has been much progress on unsupervised feature selection and several hundred papers in this area have been published in the past 5 years.  The author should include more recent state-of-the-art baselines.
  b) When tuning the hyper-parameters (e.g., \lambda) for the proposed method and baseline methods (UDFS) on validation dataset, the author should list the value range used in the parameter search. Also, as unsupervised feature selection methods, using validation dataset (which has supervision labels) to choose parameters is not a fair way, as it does use supervision information.
  c) Unsupervised feature selection papers in the past 5 years typically use 6~8 datasets to demonstrate the effectiveness while this paper only shows results on 2 datasets.

Given the reasons above, this paper needs improvement in many aspects and not ready to publish in its current form.


[1] Zhu et al. Unsupervised feature selection by regularized self-representation

[2] Yang et al. Unsupervised Feature Selection Based on Reconstruction Error Minimization

[3] Li et al. Reconstruction-based Unsupervised Feature Selection: An Embedded Approach

**Experience Assessment:**

I have published in this field for several years.

**Review Assessment: Checking Correctness Of Derivations And Theory:**

I assessed the sensibility of the derivations and theory.

**Review Assessment: Checking Correctness Of Experiments:**

I assessed the sensibility of the experiments.

**Review Assessment: Thoroughness In Paper Reading:**

I read the paper at least twice and used my best judgement in assessing the paper.

---

### Official Review · AnonReviewer3 · 2019-10-31
**Official Blind Review #3**

**Rating:** 6

**Review:**

The paper is concerned with unsupervised feature selection, in a lossy compression perspective. Formally, the idea is to select a subset of features that supports the reconstruction of the whole dataset.

The idea is good, but the the same idea (same reconstruction goal, also relying on auto-encoders) was published this year, see here: https://ecmlpkdd2019.org/programme/awards/

Algorithmically speaking, the approach is very close to the above paper; as far as I can tell, the main difference lies in the recursive feature elimination heuristics.

I feel that the originality of the paper is thus severely undermined. The authors might want to address this, through:
* thorough empirical comparisons (experimental setting, considering other datasets; comparing with other regularization schemes)
* examining the stability of the selected features (among runs).

The analysis definitely is a good point of the paper; however, parts of it are straightforward (Thm1).

Details: continuity, p.3


**Experience Assessment:**

I have published in this field for several years.

**Review Assessment: Checking Correctness Of Derivations And Theory:**

I assessed the sensibility of the derivations and theory.

**Review Assessment: Checking Correctness Of Experiments:**

I assessed the sensibility of the experiments.

**Review Assessment: Thoroughness In Paper Reading:**

I read the paper thoroughly.

---

### Decision · Program_Chairs · 2019-12-19

**Decision:**

Reject

**Comment:**

This paper proposes Restricted AutoEncoders (REAs) for unsupervised feature selection, and applies and evaluates it in applications in biology. The paper was reviewed by three experts. R1 recommends Weak Reject, identifying some specific technical concerns as well as questions about missing and unclear experimental details. R2 recommends Reject, with concerns about limited novelty and unconvincing experimental results. R3 recommends Weak Accept saying that the overall idea is good, but also feels the contribution is "severely undermined" by a recently-published paper that proposes a very similar approach. Given that that paper (at ECMLPKDD 2019) was presented just one week before the deadline for ICLR, we would not have expected the authors to cite the paper. Nevertheless, given the concerns expressed by the other reviewers and the lack of an author response to help clarify the novelty, technical concerns, and missing details, we are not able to recommend acceptance. We believe the paper does have significant merit and hope that the reviewer comments will help authors in preparing a revision for another venue.